# The Meaning of Ageing and the Educational Intervention “Good Life in Old Age”: An Ethnographic Study Reflecting the Perspective of Older Adults with Mild Intellectual Disability

**DOI:** 10.3390/ijerph22010115

**Published:** 2025-01-16

**Authors:** Marianne Holmgren, Gerd Ahlström

**Affiliations:** Department of Health Sciences, Faculty of Medicine, Lund University, P.O. Box 117, SE-221 00 Lund, Sweden; marianne.holmgren@med.lu.se

**Keywords:** ageing, education, ethnographic study, learning disability, developmental disability, older people, old age, qualitative design, well-being, healthy ageing

## Abstract

Older adults with intellectual disabilities are not adequately prepared for ageing and show anxiety and uncertainty regarding the future. Therefore, the two-year educational intervention “Good Life in Old Age” was implemented to improve their understanding of ageing and enhance their well-being. This study aimed to explore the meaning of ageing during and after the intervention from the perspective of older adults with mild intellectual disability. The ethnographic design included participant observations, field notes, group interviews, and individual follow-up interviews with 20 adults aged 44–75 (mean 63.2) with intellectual disabilities. The main findings are expressed in four themes; Awareness of ageing with intellectual disabilities, Strengthened as a person through empowering community, Awareness of vulnerability as an older adult, and The educational intervention as a resource to manage vulnerability. The education programme created a social network for healthy ageing with an atmosphere of mutual support fostering greater mental strength and self-confidence. Individual retirement plans should be created to foster socialisation, involving adapted activities and conversations about bereavement and death. There is a need to disseminate and continue developing promising education programmes for older adults with intellectual disabilities to reduce their anxiety about retirement and loneliness and facilitate healthy ageing.

## 1. Introduction

Research has shown that older adults with intellectual disabilities are not adequately prepared for the changes associated with ageing and the problems that arise in later life [1,2,3,4]. An interview study involving people aged 40–50 years revealed that they experienced considerable unspecified anxiety about ageing, as well as fear of death and uncertainty regarding the future [5]. A systematic review by Schepens et al. [6] found that few working people ≥ 50 years of age with intellectual disabilities knew what would happen after retirement, but they were worried about the loss of companions. It is well known from the literature that older people with intellectual disabilities have a more limited social network than older people without such disabilities [6,7]. They most often lack a spouse or children and, if either of their parents is alive, he or she is often very old and needs care. Few people with intellectual disabilities have ever had paid employment, which means that they do not have previous fellow workers as part of a social network after retirement. Many depend on formal support for everyday living and live in supported accommodation, such as group housing [7]. Social isolation can be a major barrier to these people’s ability to recognise and cope with the changes brought about by ageing, thereby serving as a major barrier to their healthy ageing. People with intellectual disabilities are living to an older age, with more frequent multimorbidity at an earlier age, and this is an additional reason for their being more vulnerable than older adults without intellectual disabilities [8,9,10,11].

Due to this vulnerability, attention needs to be given to healthy ageing interventions to prepare older adults with intellectual disabilities to understand the ageing process [12]. People in the general population learn about ageing from a larger network, including family, friends, and the media [12]. However, one recently published systematic review [13] and one ongoing review [14] have indicated the scarcity of interventions targeting healthy ageing, highlighting a knowledge gap regarding healthy ageing among older adults with intellectual disabilities.

To combat this knowledge gap, an educational intervention named “Good Life in Old Age” was developed and implemented [15]. The main finding of the interview study, conducted before the educational intervention, was that thoughts about ageing were avoided. The people interviewed perceived ageing as starting with retirement and were mostly worried about loneliness and boredom due to social isolation after retirement [15]. A follow-up study from the educators’ perspectives indicated that they perceived social togetherness and collaborative learning as facilitating strategies concerning the ageing process for older adults with intellectual disabilities [16]. In line with that follow-up study, this study aimed to explore the meaning of ageing during and after a two-year educational intervention from the perspective of older adults with mild intellectual disabilities.

## 2. Method

### 2.1. Design

Ethnography involves an interpretive qualitative approach that seeks to understand human beings in a particular culture through a process of “thick” or “rich” description of their everyday lives [17]. It is a question of “learning about people by learning from people” [18], learning about the relationship between people, how they interact, and their attitudes to things such as health and illness. This research involves an interplay between the individual (emic) and societal (etic) perspectives [17]. The ethnographic approach in this study was applied through participant observations, field notes, and group and individual interviews. The multiple data collection methods were applied in ethnography to achieve as rich and detailed a description as possible of the culture and its members [17].

### 2.2. The Educational Intervention

This study is the third sub-study concerning the healthy ageing intervention “Good Life in Old Age” (described in detail in [15,16]). Responsible for initiating the intervention and its management were an experienced former administrator in ageing issues for people with intellectual disabilities and a researcher and teacher in special education (not the authors). In designing the intervention, members from the local branches of the National Association for People with Intellectual Disability (FUB) gave preferences on ageing. This public involvement was carried out through meetings between the management and stakeholders, and some of them also participated in one informal interview. This developing process resulted in an education of four themes: ageing, accommodation, activity, and participation.

The goal of the educational intervention was for older adults with mild intellectual disabilities to receive support for their own ageing and become experts in healthy ageing issues, ready to share their knowledge with politicians, civil servants, journalists, and staff in disability services.

The intervention was carried out weekly between August 2021 and May 2023 in four towns across Sweden. Two towns have more than 100,000 citizens, and two have less than 100,000 citizens. Eligible for participation in the intervention were people with mild intellectual disabilities who lived in or close to one of the four towns and were well known to study circle leaders from the Adult Educational Association (Vuxenskolan) and/or the local branch of the Swedish National Association for People with Intellectual Disability (FUB). Intellectual disability means that the disability is clinically recognised during early infancy and childhood and diagnosed before age 22 [19,20]. Diagnoses are divided into mild, moderate, severe, or profound. Intellectual disabilities are characterised by significant limitations in intellectual functioning (learning, problem-solving, and judgement) and adaptive behaviour in activities in daily life, such as communication skills and social participation. People with mild intellectual disabilities have an intelligence quotient (IQ) of less than 70 but more than 50, meaning they have more abilities than people with moderate, severe, or profound intellectual disabilities [21,22].

A total of 50 meetings across four terms in each town focused on ageing, accommodation, activity, and participation [16]. The first term included 15 meetings, which started with an introduction about ageing, dementia, and lifestyle and continued with meetings about accommodations. Study visits were performed in different accommodations in their neighbourhood for people with and without intellectual disabilities. The second term also included 15 meetings, starting with a repetition from the first term, followed by decision-making meetings. Study visits concerning activities for pensioners with and without intellectual disabilities were on the agenda. The third term included 10 meetings, which started with a repetition of activities from the last term and was followed by meetings with decision makers. The follow-up meetings during the third term concerned social networks and talking about death and bereavement. The fourth and last term included 10 meetings and started with summing up terms one to three. The fourth term focused on participation, including the influence on decision making and the possibility of deepening knowledge of interesting subjects. A detailed description is published in [16].

### 2.3. The Ethnographic Research

#### 2.3.1. Sampling and Participants

The inclusion criterion for this study was that the participant should be a person with a mild intellectual disability who had participated in the two-year educational intervention “Good Life in Old Age” and agreed to participate in the participant observations, the group interviews, and the individual follow-up interview. The authors endeavoured to ensure participation was based on fully informed consent. Therefore, the presumptive participants were informed about the study by the study circle leaders or care staff in the particular town at a group meeting or individually by personal contact before the start of the educational intervention. A pre-recorded video was used, where an older adult with a mild intellectual disability was seen interviewing the authors about the aim of the research and execution of the study. The older adults interested in participating in the study were also provided with easy-to-read written information and had the opportunity to ask questions about the study.

Twenty-six people participated in the interview before the education intervention, and information on these participants was provided in the previous study [15]. One person died after the start of the study, and five people chose to prioritise other activities and, therefore, dropped out of this follow-up study. Thus, there were 20 participants in this follow-up study. The distribution among the four towns was seven, five, five, and three. There were 13 females and 7 males aged 44–75 (mean 63.2). Nine worked at the daily activities centre, two were under 65 but did not work, and nine were retired. Seven lived in their flat, one lived in their parent’s house, and twelve lived in group or service housing. (Group housing: each person has a small flat, living with 5–8 other people in the same residential area. Service housing: each person has a small flat with access to supporting staff around the clock). All participants received help from another person handling their finances, and 12 received support around the clock from disability services.

#### 2.3.2. Data Collection

The data collection was carried out from October 2021 to April 2023. Before each data collection, it was emphasised that participation was voluntary and could be cancelled at any time without giving a reason. Participant observations and field notes were noted from each of the four intervention terms, group interviews were conducted over three terms, and individual interviews were conducted post-intervention, accompanied by visualisation photos for ageing, and preceded by a questionnaire.

##### The Participant Observations, Field Notes, and Group Interviews During the Education Intervention

The authors used a manual for participant observations to better understand what happened in the meetings, what sort of atmosphere there was, and what changes occurred over time. There were 45 h of participant observations during the two-year intervention, spread over 15 days. The field notes were produced soon after the observations.

Three group interviews took place in each town after each participant observation (a total of 12 group interviews). The semi-structured interview guide was designed to elicit the participants’ perspectives regarding the content of the educational meeting. The questions were:What do you think was good about this meeting?What was it like talking about (today’s topic)?How would you describe the atmosphere in the group?What do you think of this term’s study visits?What, if anything, did you miss from the meeting today?

The group interviews lasted between 17 and 52 min (median 36.5). All interviews were digitally recorded and transcribed verbatim.

##### The Individual Interviews After the Education Intervention

Nineteen individual interviews were conducted face-to-face, and one was conducted via a computer (Zoom). These mixed-method interviews were conducted after the last educational meetings.

Firstly, the participants were asked six structured questions about their experience regarding the education content (Table 1).

After each structured interview question, the participants were asked follow-up questions to obtain detailed responses.

Secondly, the same photos as those shown in the interview before the education intervention, representing older adults in everyday situations, were shown again to facilitate remembrance of the intervention content [15]. One new photo was added showing the “stages of life”, a pictorial representation of human life as a series of ascending and descending steps.

Thirdly, a semi-structured interview was conducted to capture the participants’ educational experiences. It included the following questions:

##### Ageing

(1)Where are you in these stages of life? (Showing a picture of the stages)(2)What do you think about the fact that you are in that stage?(3)What is good ageing for you?(4)What could be better for you when you are ageing?(5)What will it be (or was it) like for you to retire? How will it be for you in terms of friends?

##### Activities for Older Adults

(6)What do you think are good activities for older adults?(7)What activities would you like to do that you do not do now?

##### Accommodation

(8)How do you want to live when you get really old? (Follow-up question: is that possible for you?) (If not, why not?)

##### The Opportunity to Influence

(9)What people have you met during the education intervention who make decisions for older adults?(10)How can you help other people who are ageing? (Follow-up question: what knowledge has the education intervention provided you with regarding how to help others with intellectual disabilities who are ageing?)

The authors interviewed the same people with intellectual disabilities in the same two towns as before the intervention. The three steps of the individual interviews, including the follow-up questions to the six structured questions, were digitally recorded and lasted between 31 and 92 min (median 53, mean 56). They were transcribed verbatim.

#### 2.3.3. Data Analysis Within the Ethnographic Approach

An abductive approach was used, which involved both inductive and deductive analysis [23,24,25]. First, both authors listened to the group and individual interviews to obtain a sense of the whole content. Parallel to listening, the authors wrote memos to facilitate the naïve understanding of the meaning of the content. Each author’s naïve understanding of the memos was thereafter discussed, and a consensus was reached regarding the use of tentative concepts that had emerged in the previous ethnographic study on the educators’ perspectives of implementing this educational intervention [16].

Then, the authors read the transcribed text as a whole to obtain an increased understanding of the content. The field notes from the participant observations and the transcribed semi-structured and group and individual interviews were integrated into one text before coding using NVivo. This provided a more complete picture of the data while making it easier to identify differences between the data sources in the analysis.

The first author conducted the coding of the interviews with the aid of the 14th version of NVivo 14 software [26]. In this deductive part of the analysis, informed by the previously identified concepts from the educators’ perspectives [16], the first author maintained an open-minded approach about the degree to which the text diverged from these concepts, which, following the deductive analysis, underwent inductive analysis.

The co-author took part in the analytical process by independently reading the text, critically reviewing the coding, and interpreting the findings. The two authors had several meetings about the similarities and differences in their interpretations until a satisfactory consensus was reached. The author’s interpretations and discussions led to the tentative concepts from the previous study, with the educators reformulated to align with the inductive content, which constituted a significant component of the analysis. The final concepts were formulated independently after the authors read the text forth and back, and the interpretations of the meaning of certain parts and the whole text were discussed at additional meetings in an iterative way before the final result emerged. The names attached to the quotations in the results section are fictitious.

## 3. Results

The experience of ageing among older adults with mild intellectual disabilities is expressed into four themes: awareness of ageing with intellectual disability, strengthened as a person through empowering community, awareness of vulnerability as an older adult, and educational intervention as a resource to manage vulnerability.

### 3.1. Awareness of Ageing with Intellectual Disabilities

The participants see ageing in a positive light. They are at home speaking about it and they are aware that it means that changes have to be made. Becoming a pensioner is a watershed. It may mean having to move, or it may mean having to cut down on or give up certain activities. Being in good health is the main thing when it comes to having a good life as an older adult. You need energy just as much as you need friends, so you do not become isolated. The participants said that they appreciated talking about ageing. Though they may not remember everything said, they left with a better understanding of their ageing and other people’s.


*I: What do you think having a good life as an older adult comes down to?*



*Well, I suppose you might say it’s a question of being healthy and keeping a sense of humour and having something to hope for, having friends around you (both old ones and new ones).*
(Isabella).


*Getting old—yes, we’ve talked a lot about that on the education. We’ve talked about when you pass away and we’ve talked about what you can and can’t do when you get to 80, 90… So, yes, I do think I’ve got something out of this education—I know things I wouldn’t have known otherwise.*
(Joe).

The participants were glad to have the chance to talk about what activities were possible for older adults, and one thing that became clear during the education intervention was that such possibilities are very limited for older adults with intellectual disabilities. The participants who were already retired were aware of this problem, but those who were younger had not thought about it before. The participants often mentioned in the interviews that there were very few activities adapted to the needs of older adults with intellectual disabilities and very few associations that meet such needs. However, study visits to well-established pensioners’ associations inspired them to establish their own. Some of the participants had tried being a member of a pensioners’ association (not adapted to people with intellectual disabilities) but had found it impossible to take part on equal terms because it was assumed, for instance, that everyone at a meeting could grasp the content of a lengthy text. In their experience, no allowance was made for intellectual disabilities. During the education intervention, the participants were made aware that activities for older adults with intellectual disabilities are at least beginning to appear in the form of get-togethers and different types of daily occupations. However, such activities are still few and far between.


*I: Activities have been a subject of discussion. Is there anything you’ve found especially useful in that respect?*



*I suppose I’d say it’s quite simply coming here and taking the education. Otherwise there’s nothing, there are no activities for older adults. You can go to the recreation centre but I don’t feel at home there. There’s nothing for you if you’re older.*
(Emily).


*I: How do you feel when it comes to the talk there’s been about activities on the education?*



*It was good. Let’s hope it’ll wake up the authorities, which nothing else will [laughs a little]. Yes, it was good—we’ve got something out of this education.*
(Tracy).

During the education intervention, the participants have been able to talk about their lives, beginning with their upbringing. There emerged a sorrowfulness deriving from a previously suppressed need to talk about sad events in the past. Furthermore, the participants appreciated being given the opportunity to talk about death. They spoke of their own experience of grief over the death of a loved one, and they listened to what others had to say about such an experience.

The study circle leaders asked the participants about how they thought the theme of death should be handled in education. One suggestion was to bring in a deacon. There was a general acceptance and great understanding of one another’s fears concerning death, which varied in strength from one participant to another. Death had been present in all participants’ lives; during the education intervention, grief and death were discussed on more occasions than had been planned. Although participants had experienced grief over someone’s death before, they had not felt able to talk about it, with death being a subject difficult to approach and, therefore, commonly avoided.


*I can cope with it better now, thanks to the education. I’ve always had difficulty when it comes to death and so on, though I do talk to the staff a lot about such things.*
(Emily).


*Something I think we ought to talk a lot more about is death. It hit me really hard when my mum died. The staff were really angry with me for being so sad about it.*
(From group interview, Town 4).

### 3.2. Strengthened as a Person Through Empowering Community

The education intervention was appreciated since it gave the participants a chance to get out and meet other people. Most group members knew one another from before, some since their school days. The intervention offered an opportunity to strengthen this acquaintanceship, and some groups became so close that they hoped to continue meeting after the intervention was over. The education intervention was a positive feature of their everyday life, affording them a welcome opportunity to get together and talk. The fact that they most often knew one another gave them a sense of security; they respected and were not afraid to say what was on their minds. This came out clearly during the periods of observation. However, a lack of tolerance could be observed in a few participants on very few occasions, which is attributable to the variation in intellectual capacity within the group. These few participants spoke of the importance of the education leader’s role in enabling everyone to speak up.


*Well, I think it’s because it’s such a little group that you get the chance to get to know everybody. If there’d been 10, 11, 12 you’d perhaps have stuck together with just one other person. When there are so few, though, this doesn’t happen. It doesn’t matter who’s present, you feel secure anyway.*
(From a group interview, Town 3).


*I: What do you think has been good about the get-togethers?*



*Participant 1: It’s interesting and you listen to one another. Yes, and it does you good to hear what others have to say.*



*Participant 2: Well, yes, I think it was an advantage that we knew one another from before, at least most of us. Another thing that made things easier was that we’re all a bit nutty—nutty in a nice way.*
(From a group interview, Town 4).

The participants talked a lot about the study visits and were strengthened by planning and discussing questions with the others in the group before and after the visits. Before each study visit, the participants formulated any questions they wanted to ask. The group provided them with the sense of security required for asking these questions. The study visits to different housing and day centres were an appreciated feature of the education intervention. It was observed that the groups were generally greatly involved and active in devising and formulating questions. Certain participants’ being less involved and active was attributable to their lower functioning of intellectual disabilities. The questions and answers were discussed after the study visit, allowing the participants to recall the visit and contemplate what they had learnt. They said that the meetings related to the study visits allowed them to share their experiences, and it was from the preparation and carrying out of such visits, together with the briefings afterwards, that they felt they were learning.


*What I think was good was that we could visit all those places, and that—and this is important!—we had the chance to work out the questions in advance.*
(from group interview, Town 1).

It was useful for the participants to learn about the types of housing where support is available, especially the housing specifically designed for older adults with intellectual disabilities. It was one thing to learn about available sheltered housing but another to see it with their own eyes. The participants were disappointed when they could not enter any flats at a particular location because all were occupied.


*Participant 1: Yes, I think the study visits were a good idea, I’ve learnt a lot from them. I don’t know whether I’d ever have set foot in those places if it hadn’t been for the education. I’d never imagined you’d be able to make those visits. It’s only if you’re in a group, isn’t it…. Yes, that was much appreciated.*



*Participant 2: We’ve gone on study visits to housing for older adults, and the last one was to housing specially designed for older adults with intellectual disability. It was interesting—as was the whole education. Then we were at that conference centre or whatever it’s called, and had a chance to meet politicians and all that.*
(From a group interview, Town 2).

The participants thought the staff on the study visits excelled in showing them the various possibilities regarding housing. If they needed to move, they wanted housing with good service, a good staff–resident ratio, and staff with adequate knowledge about intellectual disabilities. However, they were aware that such housing was scarce. They wanted to live with older adults with intellectual disabilities who were roughly the same age as they were, not with young adults who were livelier and had other interests.


*My last earthly years I don’t want to spend at home and have a flood of visitors in the form of home-help staff I haven’t met before. No, I don’t want that. What I want is to live somewhere where everybody’s nice to me and I’ve got company during the day.*
(Emma).

### 3.3. Awareness of Vulnerability as an Older Adult with an Intellectual Disability

The participants spoke of their awareness of not being like everybody else and not having the same rights. They felt that they did not have the same opportunities or social support as older adults without intellectual disabilities. Though the stated focus of the education intervention was ageing, the central consideration was sometimes the participants’ great vulnerability as individuals. The intervention offered an opportunity to air everyday problems that the participants associated with injustice. This provided them with the hope of having such problems solved. There was a particular sense of vulnerability and confusion (or chaos) regarding incomprehensible instructions, rules, and regulations in society and decisions concerning the restriction of social services. The lack of understanding concerning their need for assistance caused the participants to distrust people in authority. The groups had an atmosphere of mutual support, and the members could vent their feelings about societal injustices and be met with understanding. It emerged that participants needed to share their anxiety about what the future and old age might hold if they had no relatives and had to depend on a guardian.


*People in positions of authority shouldn’t just be thinking about people who are normal. We’re sort of not allowed to be a part of what’s going on in society. Oh no, those of us who have an intellectual disability aren’t allowed to say what we think, of course. Then again, there are some people with intellectual disability they mollycoddle, it’s almost like baby-talk the way they speak to them. I call that discrimination. It’s something I hate…. That’s why people with intellectual disability aren’t given the chance to try things out.*
(Olivia).


*I don’t know whether I feel I can rely on a guardian. I’m trying now to think in terms of having to rely on one. I feel a bit more confident about it now than I used to but it’s nothing like 99%. It’s good that there are such people, but I can’t help thinking: “What if I get swindled out of my money?” Who is there to make sure it doesn’t happen? So there’s that fear. It doesn’t go away, so I’m worried about having to see that I get a guardian. It’ll have to be a really good person that I feel secure with.*
(Samantha).


*I think that when it comes to the mobility service there shouldn’t be any letter-writing involved. I’d like it to be like it used to be, where you just phoned and spoke to somebody in the municipality. I want them to get rid of all these rules, I want to talk to someone who’s going to ask how I’m feeling.*
(From a group interview, Town 2).

### 3.4. The Educational Intervention as a Resource to Manage Vulnerability

The interviews revealed that the education intervention has provided the participants with greater mental strength, self-image, and self-confidence, thereby enhancing their capacity to combat the injustices to which they feel they are exposed. Through becoming more aware of their various problems, they have developed their ability to make their voice heard and a will to exercise influence. The intervention has given them the opportunity to describe their everyday lives to politicians, which they hope will lead to a better understanding of their needs. Meeting politicians was a much-appreciated part of the intervention, and the participants would like to repeat the experience to hammer home their needs. The politicians made promises, but the participants felt nothing happened. However, older adults with intellectual disabilities must be listened to in society.


*This education has made me tougher, more prepared to say what I think.*
(Olivia).


*I: Is there any part of the education you’d like to mention as being the most useful to you?*



*Yes, having politicians and the woman from the recreation department come here. But the main thing I’d single out is being able to make your voice heard.*
(From a group interview, Town 2).

In summary, the six-item questionnaire and its follow-up interview questions confirmed how valuable the education intervention and the companionship between the participants were. Participants said they were happy and proud to be part of the intervention. It was appreciated as an activity in its own right, and learning about ageing was meaningful to them. They looked forward to it every week and did not want it to end. They perceived it as stimulating to meet people from different organisations with different experiences and to see alternatives for accommodation and leisure activities. Some participants could express the value it had for them more clearly than others, and they felt they were being strengthened as human beings. Also, in evaluating the intervention, the mean values of the answers revealed how much they appreciated it. The highest rating (4.55) pertained to education as a whole, while the lowest (3.9) referred to housing. The participants expressed disappointment concerning the limited housing options for older adults with intellectual disabilities. The participants felt positive about sharing their newly acquired knowledge with others (like themselves with intellectual disabilities, staff, people in authority, and politicians) but felt uncertain about their abilities. Their uncertainty concerned their ability and/or whether they had the time and energy to fulfil the expert role. There was also uncertainty about how this new knowledge should be utilised. They thought they could pass on such knowledge if they were accompanied by one or more others from their group or by a member of staff.


*I: In what way would you like to pass on the knowledge and experience you’ve got from the education?*



*Well, I’d say I’d been on a education and I’d say what I’d learnt from it. I feel very proud, I really do,… being part of a group. But I wouldn’t like to sit and talk to politicians on my own, it’d be too much of a responsibility.*
(Liz).


*I’ve thought about it a lot. We’re meant to pass on what we’ve learnt to others. One thing I think is important is that it was decided you mustn’t go out on your own, there need to be two of you, and the other person should be someone who doesn’t have an intellectual disability.*
(From a group interview, Town 4).


*I think I’d be able to pass on to others what I’ve learnt from all this, I really do. A bit of it anyway. Yes, I could give tips about activities, for instance. Yes, I think so. I’m not saying I could tell them all they need to know, but something.*
(Samantha).

The participants would very much like the education intervention to be extended. This would provide them with an activity and enable them to learn more about ageing and related topics of interest.

## 4. Discussion

The experience of ageing among older adults with mild intellectual disabilities is expressed into four themes: awareness of ageing with intellectual disabilities, strengthened as a person through empowering community, awareness of vulnerability as an older adult, and educational intervention as a resource to manage vulnerability. For the participants, healthy ageing is characterised by the absence of illness, having energy, having friends, and avoiding isolation. The participants appreciated the education intervention, and the study visits to accommodation and meeting places for older adults with and without intellectual disabilities. The participants stated that the empowering community and the respectful group climate strengthened them as human beings. The preparation for the study visits made them secure and gave them the courage to ask their questions to obtain more information. The participants expressed injustice and that they did not have the same opportunities or social support from a social network as older adults without intellectual disabilities. They valued that the intervention provided them with strength and confidence, enhancing their capacity to combat the injustices they felt exposed to.

Since this study is one of three sub-studies in the project “Good Life in Old Age”, Figure 1 presents an overview of the main project’s main outcomes.

This study revealed that the participants could easily speak about ageing after the intervention and were aware that ageing concerned themselves. When interviewed before the intervention, they avoided thinking about their ageing or had a darker view of ageing. Retirement was seen as the starting point of transition to old age both before and after the intervention and was associated with stronger feelings of loneliness, social isolation, need for help in everyday life, worsening health, and death [15]. The educator’s implementation of strategies of social togetherness and learning together facilitated the processing of ageing-associated changes [16]; in this study, participants became more aware of the ageing process and hopeful about their ageing (Figure 1). These results indicate that the participants learned about the issues associated with healthy ageing. Previous studies have shown that most people with intellectual disabilities can acquire important health knowledge [13,27], despite difficulties in understanding abstract phenomena [19,28].

The positive changes in perceptions of ageing among the participants in this study confirm the need for long-term support in understanding ageing, particularly for people with cognitive impairment and a weak social network. Those in the general population learn about the ageing process from family, friends, and the media without the need for special education [12]. A greater amount of more easily accessible information on ageing in the media would facilitate learning and provide more equality for people with intellectual disabilities. Studies have found that disability service staff can have difficulty in supporting ageing people since they experience ambivalence, uncertainty, contradictions, and inconsistencies and fail to grasp both the distinctions of ageing in older adults with and without intellectual disabilities and changes in intellectual disabilities [29,30]. Furthermore, staff lack clear guidelines and strategies for supporting those who are ageing [31,32]. This is reflected in an unclear service organisation, where neither the disability service nor appropriate care services address the needs of older adults with intellectual disabilities [7] and the issue of these people’s healthy ageing. Hussain et al. [33] have proposed integrated services to reduce the existing separate sectors (“silos”) between the disability and care sectors for older adults with intellectual disabilities. One improvement would be to educate disability service staff about ageing. This would enable them to understand ageing people and provide them with knowledge-based support.

Leisure activities were highlighted as very important by participants before [15] and after the educational intervention. This is in line with the existing literature, where the role of such activities in promoting healthy ageing is also emphasised by supporters and carers (e.g., family, support workers, and housing providers) [6,34]. Several studies have emphasised engaging in social activities to feel capable and included through participation and contribution [4,5,35]. There is a risk of older adults stopping their activities after retirement when they become more isolated [36]. Therefore, Granö et al. [37] have highlighted the need to plan for retirement. They found that if staff and older adults had an plan for retirement, the older adults could continue to have a meaningful life after they have stopped working and certain daytime activities. Proactive planning ensures awareness of the person’s wishes. The participants in this study were increasingly aware of their limited options to engage in leisure activities after retirement alongside older adults without intellectual disabilities. A retirement plan initiated by staff for everyday life after retirement can help with isolation, stigmatisation, inequity, and ageism, which are risk factors for older adults with intellectual disabilities [5,38,39].

Grief and death were raised in this study as subjects for meaningful conversation, as there are usually no other forums for such conversation. The need to talk about grief and death was brought up before the intervention [15] and addressed from the educator’s perspective [16]. Previous research emphasises the importance of talking about grief and death from the perspective of older adults with intellectual disability and staff [40,41,42,43,44]. Fernández-Avalos et al. [41] established an educational intervention to deal with death and bereavement, which resulted in considerably increased understanding and a healthier attitude towards these matters. In addition, a longitudinal study of emotional impact revealed that older adults with intellectual disabilities can safely engage in end-of-life conversations/activities [42]. They found that the participants needed to talk about the end of life and were usually comfortable, but care staff needed to be prepared with strategies to provide emotional support if needed. However, staff have difficulty talking about death and dying and avoid the subject, and, therefore, they need help in overcoming their fear and anxiety in this respect. An evaluation of a training course for staff [43] showed that sharing experiences with other staff in a panel discussion and listening to older adults with intellectual disabilities tell their death-related stories were the most useful aspects of the course. This study and previous research indicate the need to discuss grief and death, suggesting that establishing groups comprising staff and older adults with intellectual disabilities, alongside the inclusion of a priest, deacon, psychologist, or equivalent, is beneficial.

The participants expressed how they experienced injustice and inequality in society in a way that older adults without an intellectual disability do not. A plausible explanation for this is stigma, which is a well-known obstacle for people with intellectual disabilities [45,46]. Negative attitudes and stereotyping of a group in society lead to prejudice and discrimination and hinder participation and accessibility for people with intellectual disabilities [46,47,48]. However, the intervention provided the participants with strength and confidence and enhanced their capacity to combat the injustices they felt they were exposed to. Disability from an early age means inequality in society and health. This inequality can be seen to increase with age, affecting healthy ageing, and, therefore, particular attention must be directed toward those who are most disadvantaged [49,50]. There is a lack of support for older adults with intellectual disabilities when it comes to making difficult decisions and life choices involving legal matters, concerning, for instance, housing and work [51]. One way to significantly improve this situation would be to invite legal experts to attend an education intervention.

From our results concerning empowering community, we can conclude that the intervention established a self-advocacy group for the participants. Self-advocacy groups have previously been described as a space where one can promote one’s own life and, thus, be self-empowered [52,53], and involve speaking up for yourself and being independent [54]. A systematic literature review by Tilley et al. [55] has shown that participation in self-advocacy groups can increase health and well-being in people with intellectual disabilities by creating new and supportive social networks and meaningful activities. Self-advocacy can also improve these people’s resources, increasing their self-esteem and self-confidence. Self-advocacy groups can also provide an environment where people learn new skills and knowledge. However, an easily overlooked finding in this study is the importance of selecting people with similar levels of cognitive impairment to create the best possible social and learning environment. Otherwise, it becomes a big task for educators to adapt education to different intellectual levels in the same group. Furthermore, an environment where people with intellectual disabilities can acquire new knowledge and engage in social interaction is shown to be important both in our study and in a study by Andersson and Bigby [54]. Since the population with intellectual disabilities is increasing, it is of great importance that these people can meet and talk about ageing in order to reduce their anxiety about it [5]. This also means they can maintain a social network, which they worry about losing as they age [6]. The findings in this study revealed that a self-advocacy group for older adults with intellectual disabilities is a good forum to meet and learn more about ageing together with other ageing people, which can boost self-confidence, reduce anxiety around ageing, and decrease instances of isolation.

Andersson and Bigby [54] found that the participants were strengthened by speaking in front of others who listened to what they had to say. In this study, the participants indicated that the intervention strengthened their self-confidence, enabling them to meet politicians and officials about their lives and ask questions they had prepared together in the groups. They wanted to share their newly acquired knowledge about ageing with those without intellectual disabilities but felt uncertain as to their ability to do so on their own [54]. Self-advocacy groups that focus on educating about ageing can serve as a model for addressing ageing concerns in the future. In developing the educational intervention pertinent to this study, older adults with intellectual disabilities have expressed their perspectives on issues important to them. This involvement process is a step forward in engaging older adults with intellectual disabilities and supporting them in disseminating knowledge about ageing to researchers and society. The development of the educational programme required many resources from management and stakeholders, as well as for the participants, who travelled away from home by transport service every week for two years. Our study shows that education interventions are promising, and, therefore, it is important to disseminate them and make them accessible to many more ageing people in disability services and care. One suggestion is to integrate education into the regular disability service, with two experienced staff members appointed as trainers to use the study guides developed in ‘Good Life in Old Age’. This would reduce the need for external resources and allow more people with intellectual disabilities to access education. To save resources, it is desirable that the training involves more than one service entity. One of the two staff should have expertise and experience caring for older people [7].

### 4.1. Methodological Considerations

There are some methodological considerations which need to be highlighted. Although the ethnographic approach was applied, with multiple data sources and participant observation over a long period [17], the authors were aware of some risks concerning the credibility of the results. Firstly, the authors used the field notes and the meetings to be self-critical regarding their role in the degree of involvement, which otherwise could influence the older adults’ behaviour. Understanding how values and attitudes influence interpersonal relationships is necessary [56]. Secondly, an effort was made to minimise differences in the results between the groups because each author was responsible for data collection in two of the four towns. This effort included our collaborative development of guides for observations, interviews, and the questionnaire for the data collection, regular discussion of the issues that appeared in the data collection, and collaboration in data analysis. Thirdly, the authors made every effort to ensure that no older adult was harmed, emotionally or otherwise. Researchers must be fully aware of their moral responsibility, and the risk to participants must be considered constantly [57,58]. Accordingly, the questions were asked at the right level for the person, we listened properly to what the person said, and we were sensitive to their wishes even if expressed in unusual words or other ways, such as through body language [59]. All data from the groups were merged to protect each person’s integrity. It should be added that the transferability of our findings is limited since the participants in this study had mild intellectual disabilities, which means that the results are not applicable to people with moderate or severe intellectual disabilities without careful consideration.

### 4.2. Conclusions and Implications

The educational intervention empowered older adults with intellectual disabilities by increasing their awareness of possibilities and challenges in ageing and the value of socialisation. The intervention gave meaning to ageing and raised awareness of the need to socialise after retirement, the need for adapted activities, and the need to talk about bereavement and death. The findings also show that the groups developed into self-advocacy groups. There is an urgent need to disseminate and continue developing these promising education interventions as support programmes for older adults with disabilities, not only to reduce their anxiety about retirement, ageing, and death but also to reduce issues of loneliness and isolation. An important implication for the future is to develop less resource-intensive educational interventions integrated into disability services to empower many more people in their ageing process and enable healthy ageing. This can include an individualised retirement plan for each older adult’s daily life, considering leisure activities and accommodation after retirement. Staff must have proper knowledge of ageing; therefore, education concerning ageing should be compulsory for people working in disability services. Future research should focus on involving older adults with intellectual disabilities as research partners to take account of their preferences and take advantage of their expertise in developing future support for older adults with intellectual disabilities.

## Figures and Tables

**Figure 1 ijerph-22-00115-f001:**
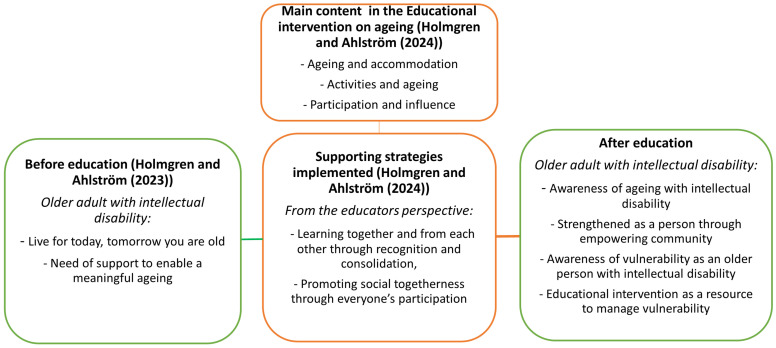
The main result expressed as themes of the three sub-studies in the project Good Life in Old Age [15,16].

**Table 1 ijerph-22-00115-t001:** Structured questions which participants were asked before individual interviews.

What did you think of the educational content as a whole?	Very good 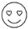 (5 points)	Good 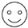 (4 points)	Neither good nor bad 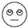 (3 points)	Bad 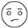 (2 points)	Very bad 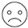 (1 point)
What did you think of the part about ageing?	Very good 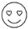 (5 points)	Good 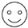 (4 points)	Neither good nor bad 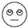 (3 points)	Bad 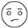 (2 points)	Very bad 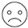 (1 point)
What did you think of the part about retiring?	Very good 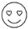 (5 points)	Good 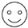 (4 points)	Neither good nor bad 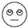 (3 points)	Bad 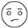 (2 points)	Very bad 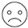 (1 point)
What did you think of the part about activities for the elderly?	Very good 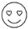 (5 points)	Good 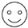 (4 points)	Neither good nor bad 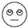 (3 points)	Bad 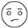 (2 points)	Very bad 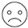 (1 point)
What did you think of the part about housing for the elderly?	Very good 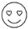 (5 points)	Good 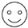 (4 points)	Neither good nor bad 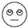 (3 points)	Bad 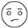 (2 points)	Very bad 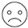 (1 point)
What opportunity do you feel you had to influence the content of the education intervention?	Very good 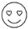 (5 points)	Good 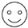 (4 points)	Neither good nor bad 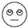 (3 points)	Bad 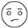 (2 points)	Very bad 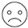 (1 point)

## Data Availability

The datasets used and analysed in this study are available upon written request from the responsible researcher (G.A.) in accordance with the ethical approval guidelines.

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
