# Peer review of "The Meaning of Ageing and the Educational Intervention “Good Life in Old Age”: An Ethnographic Study Reflecting the Perspective of Older Adults with Mild Intellectual Disability"

_ijerph, 2025, doi:10.3390/ijerph22010115_

Round 1
Reviewer 1 Report
Comments and Suggestions for Authors
Thanks for the opportunity to review this paper. Overall, the paper was well written and provided interesting data and reflections on the need for education and social support interventions for people with intellectual disability about the ageing process. The article provided some in-depth analysis of participants perspectives on what they gained from participation in the education intervention. The paper concluded with reflections on the need for further programs in this area.
The introduction was well written and provided a succinct summary and rationale for the study.
I have not read the previous papers that were part of this larger study and as such, a few more sentences during the method section may have helped to clarify areas that I was uncertain about. A description of the intervention could have a couple more sentences for those not familiar with the other references. In the results there was mention of visits etc. A bit more detail about the intervention may help the reader.
Some of the recruitment happened in a group meeting. What were these meetings about? Was this during the actual education intervention?
I was also a bit confused about the process of the individual interviews. It was mentioned that the questions were open ended but then a likert scale was used? Was the survey asked at the start and then more open ended conversation? I couldn’t follow the sequence of events of the interviews.
Also likert was incorrectly spelt as ‘lickert.’
The results section provided good in-depth analysis and my feedback is minor.
I would have liked to see the reporting of all the likert survey questions perhaps in a table.
The first theme seemed to cover both ageing and death and perhaps the heading of this theme needs to reflect that?
The first sentence of this theme reads: “The participants see ageing in a positive light.” Was that due to the education session? Did this differ from the before interviews?
What are ordinary pensioners’ associations? Perhaps explain these for an international audience.
The discussion section was a strength of the paper. The discussion was very well written and provided some interesting observations on what made this program a success and what it meant for future interventions.
Again my suggestions are minor. At times the phrasing could have been muted. For example, the sentence: “The results show that participants experience injustice and inequality in society in a way that older adults without intellectual disability do not.” Need to qualify these type of statements. The participants expressed how they experienced injustice…
‘fighting spirit’ is a cliché
I was interested in the conclusion that it was recommended that less resource intensive educational interventions should also be developed. I think more discussion about this could have been included. From the data, what are the key aspects that should be included. How could this be achieved in a less resource intensive way? Some more commentary on how to scale this program might have been interesting.
Some sentences were in the present tense rather than past tense but overall the writing was clear.
Author Response
Please, see comments in attached file.

Reviewer 2 Report
Comments and Suggestions for Authors
Overall, the authors discuss an important and underaddressed topic in the literature of people with intellectual disabilities.
As a reader, however, some of the context was difficult to understand, as findings from this study published elsewhere were not adequately covered here. Additionally, the authors discuss an ethnographic approach, but findings seem to be almost entirely from qualitative interviews and focus groups - there was not sufficient discussion of the observations in the results section to justify their inclusion in the methods - and as a reader, I would be very interested to know how those observations did or did not reflect what participants reported. Additionally, I would have liked to see more emphasis on the conclusions from this piece of the study, rather than general conclusions from previous publications. Specific recommendations:
- A copy edit is needed to address typos and grammatical issues
- Suggest moving context about previous publications from 2.2 to introduction when intervention is first introduced. Additional context is needed as well: who developed this intervention? What is the geographical and demographic scope of participation? Who was targeted and how?
- Define mild intellectual disability for a generalist audience, especially as you note that these findings would not be generalizable to people with moderate, severe, or profound intellectual disability.
- p. 3 mentions "four terms of the intervention" but this phrase is not explained, nor is the timeline of the intervention.
- Section 2.3.3 - clarify where concepts for deductive analysis came from and what they are
- Results: how do participant observations fit into your analysis?
Overall, findings show the effectiveness of the intervention, and I found the section on the development of an empowering community especially interesting - perhaps more could be said in the conclusion about the development of such communities directly addressing issues of loneliness and isolation faced by ageing people with ID?
Comments on the Quality of English LanguageTypos and some grammatical issues throughout.
Author Response
Please, see comments in attached file.
